# Is a Combination of Metals More Toxic to Mosses Than a Single Metal?

**DOI:** 10.3390/plants12233960

**Published:** 2023-11-24

**Authors:** Luigi Schillaci, Nevena Djakovic, Ingeborg Lang

**Affiliations:** Department of Functional and Evolutionary Ecology, Faculty of Life Science, University of Vienna, Djerassiplatz 1, A-1030 Vienna, Austria; luigi.schillaci@univie.ac.at (L.S.); 95nevena.d@gmail.com (N.D.)

**Keywords:** bryophytes, *Physcomitrella*, pollution, mining site, copper, manganese, iron, antimony

## Abstract

Increasing pollution in the environment calls for the precise determination of metal toxicity in plants as they are at the base of the food chain. Mosses are often employed as biomonitors and provide good models for testing metal adsorption. However, species may react differently and many studies only look at one metal at a time, even though toxicity levels are affected by metal combinations. In this study, the effects of CuCl_2_, MnCl_2_, FeCl_2_, and Sb-acetate were examined individually and in combinations on the moss species *Pohlia drummondii* and *Physcomitrium patens*. In general, the two species reacted differently to the presence of trace metals; although, for both, the tolerance limit was at 100 µM. Overall, individual metals were less toxic than combinations, with some exceptions for Fe and Mn in *P. patens*. Additionally, we demonstrate that multiple combinations of metals are especially toxic if Cu is present.

## 1. Introduction

Trace metals are elements that occur naturally in the earth’s crust and are released to the environment by various means [1]. Anthropogenic sources of trace metals are agriculture, industry, and sewage treatment plants, which contribute significantly to high metal loads in the environment [2,3]. Some trace metals, such as Cu, Fe, Mn, Zn, Ni, or Mo, are considered essential micronutrients for plants [4], animals, and humans [5]. As important as a sufficient supply of these vital trace elements is, an excess of these metals can be dangerous for living organisms [3]. Some trace metals that are not considered essential, such as Hg, Pb, or Cd, have toxic effects in all concentrations [1].

Whether a trace metal is toxic depends not only on the concentration but also on the respective effects and mechanisms that take place in living organisms. The chemical form of the trace metals and their mobility are influenced by several factors, such as the pH, the redox potential, or the water hardness [2]. In complex form, the bioavailability and, thus, the (toxic) effect of the ions are reduced. If several elements are present at the same time, either synergetic or antagonistic interactions can occur [6,7].

Bryophytes are considered to be very good bioindicators for trace metals, as well as for other xenobiotics, which some species can accumulate in high quantities [8]. There are many reasons why mosses are particularly suitable. One advantage is that they can be found in various locations, including urban and industrial areas [8]. Furthermore, they can absorb metal ions very well via the entire surface [9] and accumulate them via ion exchange sites [8,10]. In most species, the leaflets consist of a single cell layer [11] and the cuticle differs from spermatophytes in structure, chemical composition, and thickness [12]. Mosses also quickly lose water and they are therefore heavily dependent on rainfall and deposits providing them with sufficient nutrients but also potentially harmful substances. A thin film of water is often enough for the mosses to absorb dissolved substances [9]. Due to these properties, mosses have been used for decades to observe the environmental situation, in terms of pollutants or metal pollution in the air [1,13].

However, metal tolerance is species-dependent and, indeed, some mosses, such as *Mielichhoferia elongata*, need metal-rich substrates for proper development [14]. Another example is the copper moss *Scopelophila cataractae*, an obligated metallophyte. It possesses distinct physiological and developmental properties that provide phytohormone accumulation and cellular differentiation in order to profit from a high tolerance to Cu [15].

In this work, two moss species, *Pohlia drummondii* (Figure 1a) and *Physcomitrium patens* (Figure 1b), were tested for their survival in the presence of trace metals. *P. drummondii* is one of those moss species that can grow and survive in metal-rich locations and, thus, belongs to the metal-tolerant moss species [16]; however, it is also commonly found in Northern and Central Europe, North America, and Asia [17]. *P. patens* occurs naturally as an opportunist in open disturbed habitats like floodplains. It is regarded as a first-class model system for investigating developmental biology, stem cell reprogramming, and the biology of non-vascular plants [18]. Additionally, *P. patens* has been used to test metal tolerance before [1,9,19,20].

The aim was to simulate the natural habitat at a former mining site (Knappenberg; Austria) [16] and, therefore, use not only the single metals but also their combinations to understand the toxicity on mosses. Thus, we offered copper chloride (CuCl_2_), iron chloride (FeCl_2_), manganese chloride (MnCl_2_), and antimony acetate (Sb-acetate) and tested the metals individually, and in combinations, on the moss species. In particular, it was decided on to focus on combinations of two and three metals applied in solution and as a contaminated medium.

## 2. Results

### 2.1. Tolerance Limits to Single Metals in Solution

First, the toxicity tolerance of each moss species was tested individually with prepared metal solutions of CuCl_2_, FeCl_2_, MnCl_2_, and Sb-acetate at concentrations from 100 mM to 10 nM (Table 1). Three or four plantlets of the gametophore were placed in graded solutions containing one metal at a time for 48 h at room temperature. Then, the plantlets were transferred to a 0.8 M mannitol solution and the respective tolerance limits were determined by a plasmolysis test (Table 1). Plasmolysis is the detachment of the living protoplast from the cell wall. Intact membranes are a prerequisite for plasmolysis; dead cells after metal treatments cannot plasmolyze (Appendix A). So, plasmolysis can show the tolerance to a respective type and concentration of metal.

*P. drummondii* was found to be tolerant to CuCl_2_, FeCl_2_, and Sb-acetate up to a concentration of 1 mM because the detachment of the protoplast from the cell wall (plasmolysis) was clearly visible after a 48 h treatment in this concentration. For MnCl_2_, even a higher concentration of 10 mM could still be tolerated by *P. drummondii* (Table 1).

Although *P. patens* does not normally occur in trace-metal-bearing sites [21], the results for the tolerance limits of *P. patens* were comparable to the ones of *P. drummondii*. As shown in Table 1, CuCl_2_, FeCl_2_, and Sb-acetate were harmful to this species in concentrations greater than 1 mM. At 100 mM and 10 mM, the cells performed no plasmolysis. With CuCl_2_ and Sb-acetate, the chloroplasts were no longer recognizable and the color of the moss appeared gray. *P. patens* also showed a greater tolerance to MnCl_2_ when compared to the other trace metals tested but, in general, the tolerance limit is 1 mM, as it is with *P. drummondii.*

### 2.2. Tolerance Limits of Combined Metals in Solution

Both moss species had a tolerance limit of 1 mM for single metals. Assuming that metal combinations are more toxic, we repeated the tolerance test for metal combinations and offered them at concentrations of 1 mM and 100 µM (Table 2).

Of the selected trace metals, CuCl_2_, MnCl_2_, FeCl_2,_ and Sb-acetate, six solutions with combinations of two metals were prepared.

#### 2.2.1. Combinations of Two Metals

*P. drummondii* had a strong tolerance and survived in all the metal combinations by showing plasmolysis (Table 2). The combinations containing iron seemed to be the least harmful. *P. patens*, on the other hand, could not tolerate the combinations of CuCl_2_ + FeCl_2_ or CuCl_2_ + Sb-acetate at 1 mM. It also performed worse in MnCl_2_ + Sb-acetate, even at 100 µM. Although both species survived the metal combinations up to the respective concentration limits, visual observation of the moss cells showed differences in some cases, like more affected cells at the tip of stemlets or leaves.

#### 2.2.2. Combinations of Three Metals

Table 3 shows the tolerance limits of *P. drummondii* and *P. patens* to solutions containing three metals at the time, either at concentrations of 1 mM or 100 µM. The results are similar to the tests with only two metals in combination.

*P. drummondii* clearly survived the concentration of 100 µM in all metal solutions; the tolerance limit to the triple metal combination is at 1 mM. For *P. patens*, on the other hand, the combinations CuCl_2_+ MnCl_2_+ FeCl_2_ and CuCl_2_ + FeCl_2_ + Sb-acetate were the most harmful, hinting towards a toxic combination of Fe and Cu. In all other metal combinations, the tolerance limit of *P. patens* was similar to *P. drummondii*.

#### 2.2.3. Combinations of Four Metals

After 48 h in a solution containing a quadruple combination of trace metals, both samples were brown in the lower part of the leaf while the upper part still had the characteristic green color. In a solution containing a concentration of metals of 100 μM, *P. drummondii* clearly showed plasmolysis while *P. patens* was suffering (Table 4). In the solution containing a final concentration of 1 mM of all four metals, both species were affected more than they were in 100 μM; although, *P. drummondi* had more living cells than *P. patens*.

#### 2.2.4. Pohlia drummondii

In order to investigate whether and what sort of influence the selected metals, CuCl_2_, FeCl_2_, MnCl_2_, and Sb-acetate, have on the growth of the mosses, agar plates were prepared in which the metals were incorporated. For single metals, and for the combinations of metals, we used a final concentration of 100 μM because P. patens did not survive some of the triple combinations at 1 mM concentrations (Table 3).

In the single metal treatments on agar (Figure 2), P. drummondii was continuously producing biomass over the observation period of four weeks. If compared to the other metals, the growth of P. drummondii was significantly limited by CuCl_2_. In copper concentrations, there was hardly any increase in biomass over the experimental time of four weeks (Figure 2a). For FeCl_2_, there was a significant difference in growth compared to the control (Figure 2b). MnCl_2_ caused similar results to FeCl_2_ (Figure 2c). In the plate spiked with Sb-acetate, P. drummondii grew better than in the presence of the other single metals; however, growth was still much less than the control (Figure 2d).

Figure 3 shows the growth data of P. drummondii on agar plates containing two individual metals (100 µM final concentration), the combination of these two metals (100 µM final concentration), and the control. CuCl_2_ has significantly increased growth in combination with FeCl_2_ compared to these individual metals and samples grown with other double combinations of metals (Figure 3a). In contrast, if Cu is offered in combination with Mn or Sb, i.e., as CuCl_2_ + MnCl_2_ or CuCl_2_ + Sb-acetate, the combination of the two metals is never more harmful than the single presence of CuCl_2_ but has a growth similar to the single presence of Mn and Sb, respectively (Figure 3b,c). Fe also results in a slightly positive effect if offered in combination with Mn (Figure 3d). For the treatments MnCl_2_ + Sb-acetate and Sb-acetate + FeCl_2_, we found a reduction in growth by the combination of the metals and a particularly slow start in weeks 1 and 2 (Figure 3d,e).

The growth of P. drummondii plantlets in combinations of three metals within the agar is summarized in Figure 4. In this figure, we have also included the growth data on agar with individual metals, as well as double combinations of the respective metals. This way, the combinatory effect of the metals is highlighted. In Figure 4a, the combination of CuCl_2_ + MnCl_2_ + FeCl_2_ can be compared to the individual metals FeCl_2_ and MnCl_2_, as well as to the double combinations CuCl_2_ + MnCl_2_, CuCl_2_ + FeCl_2_ and MnCl_2_ + FeCl_2_ and the control, respectively; although, the most biomass was produced in the control plates without any metal (Figure 4a). A similar effect is shown for the other triple combinations, too. Triple combinations of CuCl_2_ + FeCl_2_ + Sb-acetate (Figure 4b) and CuCl_2_ + MnCl_2_ + Sb-acetate (Figure 4c), respectively, did not cause further detrimental effects but, in contrast, proved less harmful than, e.g., CuCl_2_ alone. The combination MnCl_2_ + FeCl_2_ + Sb-acetate (Figure 4d) was less harmful than double combinations containing Sb-acetate.

P. drummondii, grown on an agar containing the quadruple combination of metals is shown in Figure 5. We observed a constant increase in biomass and this suggests a lower toxicity or an excellent tolerance of this moss in the presence of four metals in combination. In almost all other settings of single, double, and triple combinations, this moss has grown well, except for in the presence of CuCl_2_ as a single metal.

#### 2.2.5. Physcomitrium patens

As for P. drummondii, we first tested the individual trace metals in concentrations of 1 mM and 100μM on P. patens (Table 3). For metal combinations in the medium, the final concentration of 100μM was used.

In the single metal treatments of P. patens on agar (Figure 6), we observed a continuous gain in biomass for all metals, except for CuCl_2_. This metal proved to be most toxic to P. patens (Figure 6a), similarly to P. drummondii (Figure 4a). In the case of FeCl_2_ (Figure 6b), the samples grew slowly and linearly. MnCl_2_ appeared to be least harmful to P. patens, with a continuous gain in biomass. At the end of the observation period, however, MnCl_2_-treated samples grew better than the control (Figure 6c). P. patens grown on 100 μM Sb-acetate had constant biomass production but lower than both the control samples and the samples grown in the presence of other metals, except for CuCl_2_. (Figure 6d).

In Figure 7, growth data on double metal combinations are summarized for P. patens. The results show plates containing the two metals separately (100 μM final concentration), in combination (100 μM final concentration), and the control. The most significant differences occur in all the combinations containing CuCl_2_ (Figure 7a–c), where biomass production is almost zero. In the double combinations without CuCl_2_, i.e., MnCl_2_ + FeCl_2_ (Figure 7d), MnCl_2_ + Sb-acetate (Figure 7e), and FeCl_2_ + Sb-acetate (Figure 7f), growth is reduced in comparison to the control.

The toxic effect of Cu was also clearly visible for P. patens when grown on agar with three metals in combination (Figure 8). The combination CuCl_2_ + MnCl_2_ + FeCl_2_ looks harmful, with values comparable to CuCl_2_, CuCl_2_ + MnCl_2_, and CuCl_2_ + FeCl_2_ (Figure 8a, Figure 7a,b). Similarly, the combinations CuCl_2_ + MnCl_2_ + Sb-acetate (Figure 8b) and CuCl_2_+ FeCl_2_ + Sb-acetate (Figure 8c) were also harmful to this moss species and growth was severely restricted. The triple metal combination without Cu, i.e., MnCl_2_ + FeCl_2_ + Sb-acetate (Figure 8d), resulted in better growth of P. patens plantlets; although, the samples did not produce as much biomass as in single or double metal combinations of Mn, Fe, or Sb-acetate, respectively, or in the control.

P. patens cultured on agar containing the four metals in combination (Figure 9) clearly showed the great toxicity of these four metals together, especially when compared to the control. There is only a poor production of biomass, similar to that found in agar containing Cu or Cu-containing combinations.

## 3. Discussion

The survival of the two moss species, P. drummondii and P. patens, under the influence of the trace metals Cu, Fe, Mn, and Sb was tested by means of plasmolytic tolerance tests and growth studies. In general, we wanted to know if the two moss species reacted differently to the applied trace metals in a short-term stress condition of four weeks because different species are used for biomonitoring [22]. More specifically, we hypothesized that metal tolerance is lower if combinations of two, three, or four metals are offered at the same time.

Mosses can absorb and accumulate trace metal via various mechanisms. After uptake, the metals can either be bonded to the cell wall and outer surface of the plasma membrane—in the exchangeable form to exchange—or chelate binding sites, accumulate as solid particles in the surface layer, or enter the cell interior where they are dissolved or undissolved [1]. The main uptake is via ion exchange processes and the formation of chelate complexes on the cell wall [1,23]. This is made possible by organic, functional groups in the cell wall, which can interact with ions and lead to ion exchange or chelation. Examples of functional groups that may be involved include carboxyl and phosphoryl groups, polyphenols, or amino groups [1,23]. Since ion exchange is a physiological-chemical process, several factors play an important role in the absorption. The climatic conditions, the pH, and the drying process are just as important as the cell age, the type of free cations, and the type of pollutant [10].

### 3.1. Investigated Metals

Cu, Fe, and Mn are essential trace elements for all living organisms [5]; only Sb-acetate is not considered a micronutrient. Cu can be present as Cu (I) or Cu (II), as well as in the form of complexes that all have different properties. Fe can be present in aqueous solutions as Fe (II) or as Fe (III); Mn is predominant in soils with an oxidation level of +2, +3, and +4 [24]; and Sb can be found in many minerals, mainly in combination with sulfur [25] or oxygen or mixed with copper, lead, or mercury sulfide.

On the surface of moss cells, the extracellular enrichment of the metals takes place via the ion exchange processes and, additionally, a complex formation with functional organic groups, especially carboxyl and phosphoryl groups, occurs [1]. However, differences in the moss species can strongly influence the reactions to pollutants, such as trace metals. Interestingly, different tolerance limits were recently correlated to cell shape and cell size by Petschinger et al. [26]. In cells with thicker cell walls, the proportion of apoplast in the total surface of the leaf is greater than in cells with thinner cell walls. Therefore, moss species with thicker cell walls were more tolerant to trace metals than species with thin cell walls, as is the case in P. patens [26].

The tolerance tests in our study clearly showed that P. drummondii was less affected than P. patens and survived better in 1 mM metal solutions (Table 1, Table 2 and Table 3).

### 3.2. Tolerance to Trace Metals

The adsorption and availability of these trace metals for plants depend mainly on the pH of the soil [24]. In the soil of the former mining site, Knappenberg, Fe is practically immobile and has minimal plant toxicity. Similarly, Mn, although present in extremely high concentrations, is hardly accessible to plants. Cu concentrations indicate high soil values and Cu carbonates have been detected significantly more commonly than sulfides due to the alkaline pH. Sb was significantly less prevalent than Cu, Fe, and Mn and was primarily adsorbed by Fe hydroxides [16]. Hence, the availability of metals and the respective metal-linked anion play a major role in the uptake and adsorption of both nutrients and pollutants, such as trace metals. In general, the metals must be present in an available form as free cations [19,21]. Solubility data from Sassmann et al. [21] showed that Cu ionic salts were weakly soluble, depending on the cation (CuSO_4_ at 0.1 mM: 17%; CuCl_2_ at 0.1 mM: 21%), and lower metal concentrations resulted in an increase in the proportion of soluble copper ions. On the other hand, in the case of EDTA chelates, the soluble EDTA complexes accounted for 99% of the total soluble metals; whereas, free metal concentrations were less than 0.28% of the total soluble metals [21].

Additionally, Petschinger et al. [26] reported a relationship between the lamina cell shape and metal tolerance in mosses. The species with long and thin lamina cells presented a greater tolerance than species that had isodiametric cells, especially if treated with iron. The thickness of the cell wall also played an essential role in the tolerance to specific metals, most likely because of its ability to absorb positively charged ions [26].

In our studies, Cu was particularly harmful and P. patens reacted with a strong reduction in growth. If Cu was offered to P. patens as chloride or sulfate or was EDTA-linked, EDTA had the lowest toxicity. The anions themselves were not toxic but contributed to the harmful effect of the metals by their ability to form chelates, thereby changing the availability of the metals [21]. Furthermore, the number of freely available metal cations in combination with chloride is higher and, thus, the metal absorption is favored in P. patens. This is a big difference to seed plants where metal uptake is favored by EDTA in the form of water-soluble metal chelates [19,21].

In contrast to the harmful effect of Cu, Mn had a positive effect on the growth of both moss species, P. drummondii and P. patens. Mn is an essential trace element and Boquete et al. (2011) found a difference between exposure and tissue concentration in moss biomonitors. The authors attributed this fact to a good recycling capacity for the absorption and concentration of this metal. In addition, Na and Mg ions can easily displace Mn from the cation exchange sites, thereby reducing the concentrations of manganese [27]. In our study, Mg was present in the agar media as MgSO_4_. It could have caused a displacement of Mn and, thus, good growth rates of the Mn samples. A combination with Cu, e.g., CuCl_2_ + MnCl_2_, however, caused growth restrictions.

Similarly, Cu in combination with Fe and Sb also reduced the growth rates considerably. Specific uptake mechanisms for Sb are not fully understood [25]; however, it is believed that Sb (III) is more toxic than Sb (V). Thus, Sb (III) is converted into the less harmful Sb (V) after uptake and stored in the vacuole or complexed with proteins. These assumptions were confirmed in a study by Diaz et al. [28]. In the aquatic moss Fontinalis antipyretica, Sb had the lowest toxicity compared to selenium and mercury. In the present work, Sb was used in the form of Sb-(III)-acetate and, therefore, we attributed the negative effect in the combination of CuCl_2_ + Sb-acetate to the influence of CuCl_2_.

It has been known for a long time that Fe accumulation in moss gametophytes is two to three times greater than the Mn and Cu concentrations and, possibly, Fe is the most important element in regulating the concentration of other metals [29]. Combinations of Cu + Fe and Fe + Mn have been shown to have synergetic effects and positive correlations were found for Fe + Sb, as well as for Mn and Sb; meanwhile, Cu + Mn could be synergetic or antagonistic [6,7]. Depending on the effect, concentration, and toxicity of the metal, these synergetic or antagonistic effects need to be judged individually for the species tested.

In triple combinations of the metals used in this study, the harmful effect of two metal combinations increased even further in P. patens; howerver, the effect was partially compensated in the moss from the metal habitat P. drummondii.

As for the quadruple combination of metals, we noticed that the tolerant species P. drummondii showed constant and linear growth over the course of four weeks. P. patens, instead, was suffering with almost zero growth.

### 3.3. Reactive Oxygen Species (ROS)

Trace metals increase the production of reactive oxygen species (ROS) in the cells [30]. In mosses, Antreich et al. [20] found differences between the tip and base of the stemlets, as well as differences according to the metal-bound anion.

Wu et al. [31] investigated ROS production after CuCl_2_ treatment in the moss Plagiomnium cuspidatum, which is, like P. patens, a deciduous species. Cu-induced H_2_O_2_ was localized in the plasma membrane and cell wall and O_2_ was mainly on the inner side of the plasma membrane and in the cytoplasm around the chloroplasts. The reactions of the ROS with lipids, proteins, and nucleic acids caused damage to the membranes or inactivation of the enzymes [31]. Increasing Cu accumulation caused a strong decrease in chlorophyll a and b content in mosses [32], thereby affecting photosynthesis. Hence, the toxicity of trace metals could be a consecutive effect of the metal treatment. On the other hand, the permeability of damaged membranes is altered and they can thus absorb more metals and bind them inside the cell to new cation exchange sites [33]. In one of the few studies that use metal combinations, Sun et al. [34] tested a combination of lead and nickel. They also reported an increase in ROS, such as H_2_O_2_, and the influence of the antioxidant defense system [34].

## 4. Materials and Methods

### 4.1. Plant Material and Trace Metals

The metals were chosen according to a former mining site and the natural habitat of P. drummondii, Knappenberg, Austria (47°42′11.23″ N, 15°47′30.04″ E, 860 m a. s. l.). At this well-studied collection site, the selected metals were the most present in the environment [16]. In the laboratory experiments, we offered them as CuCl_2_, FeCl_2_, MnCl_2_, and Sb-acetate both in solution and mixed into the solid growth medium at the final concentrations of 100 mM and 10 mM (single metals) and at the final concentration of 100 μM (metal combinations), respectively.

The selected mosses, Pohlia drummondii ((Müll.Hal.) A.L.Andrews) and Physcomitrium patens ((Hedw.) Bruch and Schimper), are kept in a sterile culture at the Department of Functional and Evolutionary Ecology, the University of Vienna, Djerassiplatz 1, A-1030 Vienna Austria. Before the experiments started, the mosses were inoculated on fresh agar plates with nutrient medium (control; https://sites.dartmouth.edu/bezanillalab/moss-methods/ (accessed on 23 October 2023)). In brief, the control medium consisted of 1.03 mM MgSO_4_, 1.86 mM KH_2_PO_4_, 3.3 mM Ca(NO_3_)_2_, 2.7 mM (NH_4_)2-tartrate, 45 μM FeSO_4_, 9.93 μM H_3_BO_3_, 220 nM CuSO_4_, 1.966 μM MnCl_2_, 231 nM CoCl_2_, 191 nM ZnSO_4_, 169 nM KI, 103 nM Na_2_MoO_4_, and 0.7% of agar. All the chemicals were purchased from Sigma Aldrich (Vienna, Austria). For the control, the medium nutrients and agar were added to distilled water, mixed, autoclaved, and poured into sterile Petri dishes (6 cm, Greiner bio-one, Kremsmünster, Austria) on a sterile bench. For the metal treatments, the respective trace elements and combinations thereof were added at a final concentration of 100 μM before autoclaving. After cooling the agar, fresh plant material was propagated in those plates. The mosses were grown over a period of four weeks under controlled conditions at 20 °C and with a 14/10 h light/dark regime.

### 4.2. Plasmolytic Tolerance Trials

All tolerance tests were performed in a minimum of two separate repeats with at least three biological replicates.

The determination of the tolerance limits of the mosses to the trace metals, CuCl_2_, FeCl_2_, MnCl_2_, and Sb-(III)-acetate, was carried out by plasmolysis studies (Appendix A). Plasmolysis occurs when plant cells are placed in hypertonic solutions: by osmotic water extraction from the vacuole, the protoplast becomes detached from the cell wall. The effect only takes place in living cells with intact, permeable membranes. Hence, plasmolysis only occurs in living cells and, as reported in other studies [26,35], is a valid method to determine the viability of a cell [36]. If 50% of cells showed plasmolysis (i.e., detachment of the plasma membrane from the cell wall), we considered it [+/−]; the categorizations were more than 50% [+] and less than 50% [−].

For the selected metals, CuCl_2_, MnCl_2_, FeCl_2,_ and Sb-(III)-acetate, stock solutions were prepared at a concentration of 1 M/L. The tolerance tests were performed in a 96-microtiter plate. The metal solutions were pipetted as a dilution series of concentrations from 100 mM to 1 nM (10^−1^ M to 10^−8^ M). Moss samples (gametophores consisting of leaflets and stems) were then placed in the solutions for 48 h at room temperature and the microtiter plate was sealed with a lid. Technical replicates of each sample were produced at least twice.

After 48 h, the moss samples from the graded metal solutions were transferred into a 0.8 M solution of mannitol (Sigma Aldrich). After removal from the metal solutions, the mosses were lightly dabbed on filter paper and immediately immersed in the mannitol solution, where they remained for 30 min.

The assessment of plasmolysis and, thus, the survival of the mosses was carried out on a light microscope (Olympus BX41) linked to a digital camera (Nikon DS-Fi3) and the corresponding program (Nikon NIS-Elements).

### 4.3. Combined Solutions

Solutions of two, three, or four combined metals were prepared to answer the question of whether metals in combination are more harmful to the mosses than alone. In total, combinations with two, three, and four metals, were prepared. The final concentrations of solutions with combined metals were always 100 µM, as well as for each individual metal. The metals were offered at equal ratios of 1:1 (two metals), 1:1:1 (three metals), and 1:1:1:1 (four metals). The procedure for the tolerance tests with combined metal solutions was the same as for single metals. All tests were carried out in 96-well plates, as described above.

### 4.4. Growth of the Mosses

In addition to the tolerance tests, the growth of the mosses was analyzed over a period of 4 weeks. The normal nutrient agar (control) was prepared and the metals were added at final concentrations of 1 mM and 100 μM (individual metals), as well as at final concentrations of 100 μM for combined metals. Fresh moss transplants were inoculated and the growth rate was documented over a period of 4 weeks. Photographic records of the plates were made at the end of each week using a digital camera (Canon EOS 2000D, Canon, Tokyo, Japan) on a fixed stage. The programs Image J/Fiji (fiji-win32) and Microsoft Excel 16.70.1 were used to graphically display the growth progress in mm^2^.

## 5. Conclusions

Although often highly sophisticated, laboratory conditions only partially reflect the situation in a natural habitat. Several characteristics are not fully reproducible, such as particular climatic conditions, sudden temperature changes, irradiation angles, long observation periods, or combinations of them. Here, we reported that the toxicity of the metal combinations simulating a natural habitat depended on the moss species and the metal concentration. P. drummondii is considered a metal-tolerant moss; whereas, P. patens is not. These characteristics were also reflected in the results. P. drummondii showed a better tolerance towards the trace metal combinations than P. patens. In our experiments, Cu was more harmful than Mn, Sb, and Fe and both moss species showed growth restriction with Cu and all combinations containing this metal. Future studies with higher special resolution, e.g., at a Synchrotron light source, might help us to understand the intracellular localization and distribution of metals in mosses.

## Figures and Tables

**Figure 1 plants-12-03960-f001:**
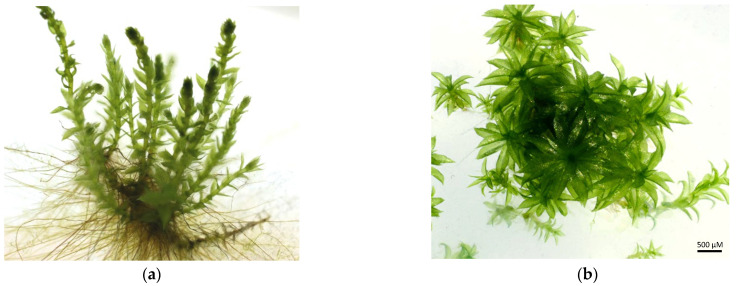
Habitus of the investigated moss species. (**a**): *P. drummondii*, (**b**): *P. patens*.

**Figure 2 plants-12-03960-f002:**
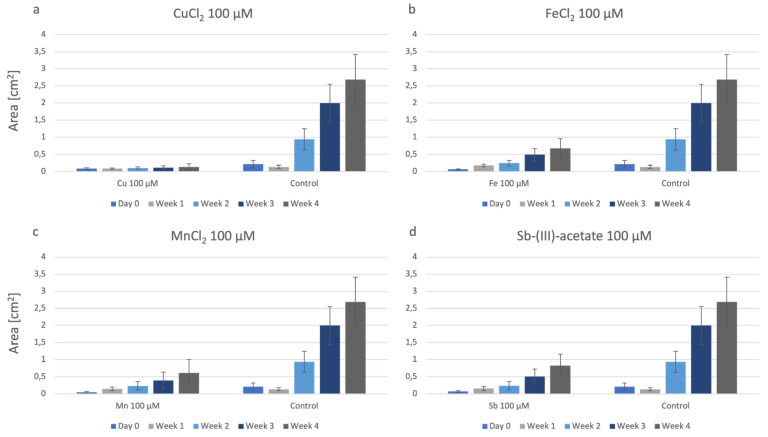
Growth of P. drummondii under the influence of: (**a**): CuCl_2_; (**b**): FeCl_2_; (**c**): MnCl_2_; (**d**): Sb-acetate.

**Figure 3 plants-12-03960-f003:**
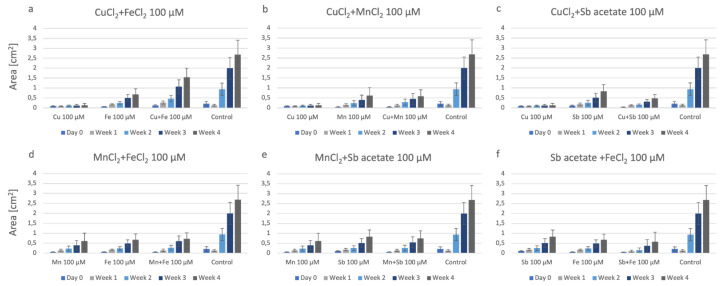
Growth of P. drummondii on agar containing combinations of two metals; for better comparison, growth data of the respective single metals and controls are given. All at 100 µM final concentrations. (**a**): CuCl_2_ + FeCl_2_; (**b**): CuCl_2_ + MnCl_2_; (**c**): CuCl_2_ + Sb-acetate; (**d**): MnCl_2_ + FeCl_2_; (**e**): MnCl_2_ + Sb-acetate; (**f**): Sb-acetate + FeCl_2_.

**Figure 4 plants-12-03960-f004:**
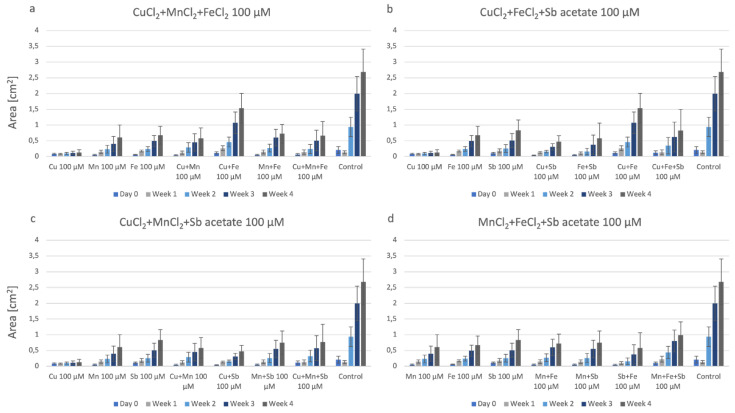
Growth of P. drummondii under the influence of triple metals. Each set also shows the data on agar containing single or double metals at 100 µM, respectively. (**a**): CuCl_2_ + MnCl_2_ + FeCl_2_; (**b**): CuCl_2_ + FeCl_2_ + Sb-acetate; (**c**): CuCl_2_ + MnCl_2_ + Sb-acetate; (**d**): MnCl_2_ + FeCl_2_ + Sb-acetate.

**Figure 5 plants-12-03960-f005:**
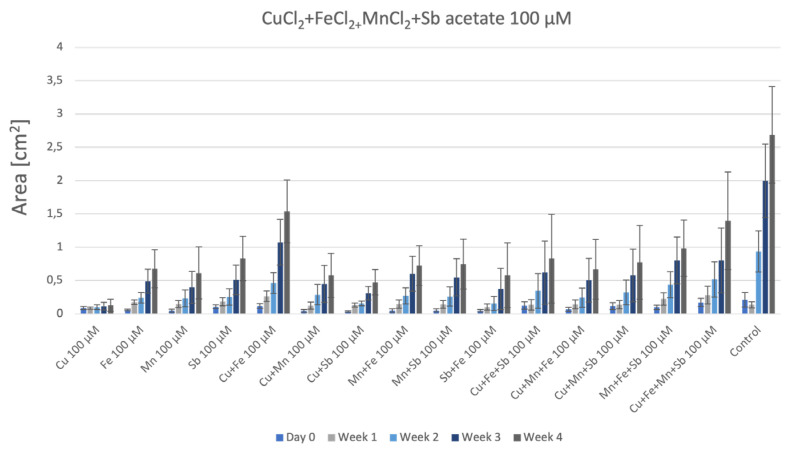
Growth of P. drummondii under the influence of quadruple metals. Each set also shows the data on agar containing single, double, or triple metals at 100 µM.

**Figure 6 plants-12-03960-f006:**
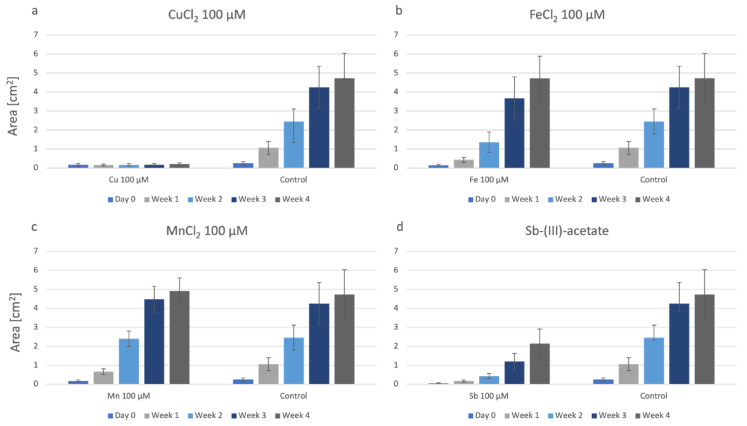
Growth of P. patens under the influence of: (**a**): CuCl_2_; (**b**): FeCl_2_; (**c**): MnCl_2_; (**d**): Sb-acetate.

**Figure 7 plants-12-03960-f007:**
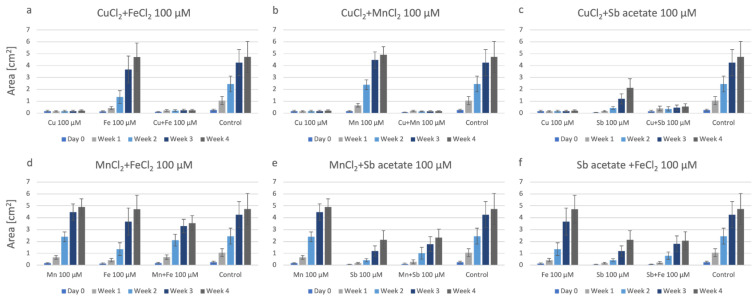
The growth of P. patens under the influence of: (**a**): CuCl_2_ + FeCl_2_; (**b**): CuCl_2_ + MnCl_2_; (**c**): CuCl_2_ + Sb-acetate; (**d**): MnCl_2_ + FeCl_2_; (**e**): MnCl_2_ + Sb-acetate; (**f**): Sb-acetate + FeCl_2_.

**Figure 8 plants-12-03960-f008:**
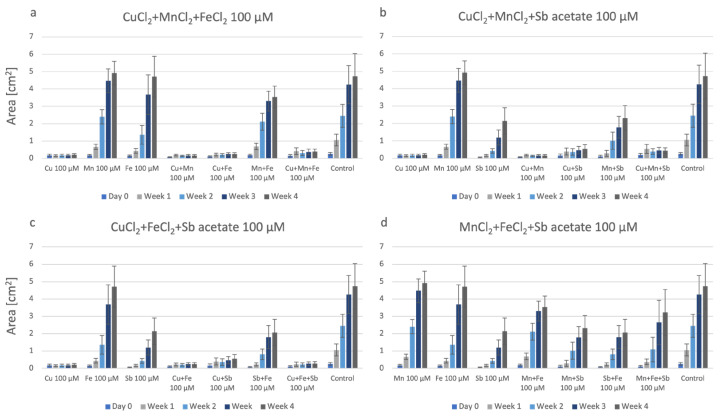
The growth of P. patens on tiple metal combinations: (**a**): CuCl_2_ + MnCl_2_ + FeCl_2_; (**b**): CuCl_2_ + MnCl_2_ + Sb-acetate; (**c**): CuCl_2_ + FeCl_2_ + Sb-acetate; (**d**): MnCl_2_ + FeCl_2_ + Sb-acetate.

**Figure 9 plants-12-03960-f009:**
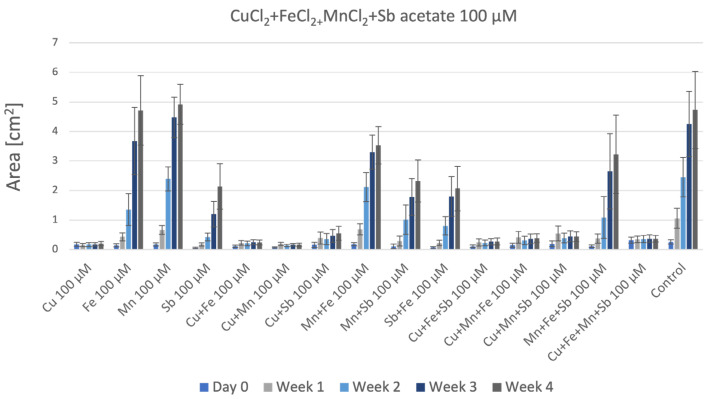
P. patens grown on four metals. Each set shows the data on agar containing single, double, triple, or quadruple metals at a 100 µM final concentration.

**Table 1 plants-12-03960-t001:** Plasmolysis assessment for single metals in solution, with concentrations from 100 mM to 10 nM. No plasmolysis (−), plasmolysis (+), tolerance limit (+/−).

Metals	Concentration in Solution							
*Pohlia drummondii*								
	100 mM	10 mM	1 mM	100 μM	10 μM	1 μM	100 mM	10 nM
CuCl_2_	−	−	+/−	+	+	+	+	+
FeCl_2_	−	−	+/−	+	+	+	+	+
MnCl_2_	−	+	+	+	+	+	+	+
Sb-acetate	−	−	+/−	+	+	+	+	+
*Physcomitrium patens*								
CuCl_2_	−	−	+/−	+	+	+	+	+
FeCl_2_	−	−	+/−	+	+	+	+	+
MnCl_2_	−	+	+	+	+	+	+	+
Sb-acetate	−	−	+/−	+	+	+	+	+

**Table 2 plants-12-03960-t002:** Plasmolysis assessment for combinations of two metals in solution, with concentrations of 1 mM and 100 μM. No plasmolysis (−), plasmolysis (+), tolerance limit (+/−).

Metals	Concentration in Solution			
	*Pohlia drummondii*		*Physcomitrium patens*	
	1 mM	100 μM	1 mM	100 μM
CuCl_2_ + FeCl_2_	+	+	−	+
CuCl_2_ + MnCl_2_	+/−	+	+/−	+
CuCl_2_ + Sb-acetate	+/−	+	−	+
MnCl_2_ + FeCl_2_	+/−	+	−	+
MnCl _2_+ Sb-acetate	+	+	+/−	+/−
FeCl_2_ + Sb-acetate	+/−	+	+/−	+

**Table 3 plants-12-03960-t003:** Plasmolysis assessment for combinations of three metals in solutions of 1 mM and 100 μM. No plasmolysis (−), plasmolysis (+), tolerance limit (+/−).

Metals	Concentration in Solution			
	*Pohlia drummondii*		*Physcomitrium patens*	
	1 mM	100 μM	1 mM	100 μM
CuCl_2_ + MnCl_2_ + FeCl_2_	+/−	+	−	+
CuCl_2_ + MnCl_2_ + Sb-acetate	+/−	+	+/−	+
CuCl_2_ + FeCl_2_ + Sb-acetate	+/−	+	−	+
MnCl_2_ + FeCl_2_ + Sb-acetate	+/−	+	+/−	+

**Table 4 plants-12-03960-t004:** Plasmolysis assessment for combinations of four metals in solutions of 1 mM and 100 μM. No plasmolysis (−), plasmolysis (+), tolerance limit (+/−).

Metals	Concentration in Solution			
	*Pohlia drummondii*		*Physcomitrium patens*	
	1 mM	100 μM	1 mM	100 μM
MnCl_2_ + FeCl_2_ + MnCl_2+_Sb-acetate	+/−	+	−	+/−

## Data Availability

All research data are stored at the repository at the University of Vienna, according to the actual data manager plan. The data are available from the authors upon request.

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
