# Peer review of "Is a Combination of Metals More Toxic to Mosses Than a Single Metal?"

_plants, 2023, doi:10.3390/plants12233960_

Round 1
Reviewer 1 Report (Previous Reviewer 1)
Comments and Suggestions for Authors
Now the paper is ok for publication
Author Response
Please see attachment

Reviewer 2 Report (Previous Reviewer 2)
Comments and Suggestions for Authors
The manuscript entitled “Are multiple combinations of metals more toxic to mosses than single metals?” aims to address if exposure to a combination of metals can be more toxic to mosses than exposure to single metals. This phenomenon is often referred to as "synergistic toxicity" or "additive toxicity".
I think this research has certain innovations. Indeed, Mosses can efficiently absorb and accumulate various metals from the atmosphere and substrate, making them excellent indicators of metal contamination. They serve as "sponges" for atmospheric pollutants, especially metals.
Further, I think that the topic content of this manuscript is also suitable for the international audiences of Plants.
I encourage the authors to review and modify carefully the manuscript and after checking the below points I am sure it should be accepted for publication:
- The use of the term "heavy metals", which was never internationally defined, should be discouraged. The term "heavy metals" should be replaced by "trace elements" (TE), "metals" "trace metals", "toxic elements" (for metal whose toxicity is proven) or "potentially toxic elements".
- Botanical descriptions are an important tool for plant identification. It provides a comprehensive overview of the plant's characteristics. It should be mentioned in the text!
- Photos in figure 1 may not adequately convey the appearance of individuals being studied.
- L52 to L 62 should be mentioned in the conclusion section.
- L 263 : « Investigated metals » this subtitle is more suitable.
- L264-268 : this paragraph should be placed in Materials and Methods section.
- Avoid putting (Fig...) in the discussion section.
- This is a short-term metal stress application and short term mosses responses. The authors in the discussion or in conclusion section must take into consideration this condition.
- The manuscript lack a paragraph to show what is the important of this study to engineers in future study?
- The manuscript is need to improving in English language, please improve it.

Moderate editing of English language is required
Author Response
Thank you
This manuscript is a resubmission of an earlier submission. The following is a list of the peer review reports and author responses from that submission.
Round 1
Reviewer 1 Report
Comments and Suggestions for Authors
The paper does not include any data for the statistical significance evaluation, and has to be rejected by this reason. Moreover the authors reported that two metals are more harmful that a single one, however they do not show how such broad conclusion comes out from results: authors found that Mn and Fe add toxitity of Cu, but Mn does not add toxitity to Fe (they simply did not study such combinations), while the title implies that the latter rule should be also valid.
What is the main question addressed by the research? - Plant (moss) tolerance to heavy metals
Is it relevant and interesting? yes
How original is the topic? moderately
What does it add to the subject area compared with other published material? - Nothing as data are poorly presented
Is the paper well written? It is well written where it is written, bit a large part whcih is necessary is not written and either not done, or results appeared so poor that author did not present them
Is the text clear and easy to read? - Clear that nothing serious is done
Are the conclusions consistent with the evidence and arguments presented? Impossible to evaluate (but must be possible, and here is the main problem of the paper)
Do they address the main question posed? yes
moderately acceptable
Reviewer 2 Report
Comments and Suggestions for Authors
Authors: Schillaci et al.,
The manuscript (plants-2541024) entitled “Are double and triple combinations of metals more toxic to 2 mosses than single metals?”. In this study, authors examined the effect of CuCl2, MnCl2, FeCl2 and Sb-acetate individually and in combinations on the moss species Pohlia drummondii and Physcomitrium patens. Results showed that globaly, the two species reacted differently to the presence of heavy metals although for both, the tolerance limit was at 100 µM for the metal combinations and even higher for individual metals. This is a relevant topic that needs to be addressed. Overall, the quality of the manuscript is acceptable, but some points need to be improved before its publication.
Minor comments.
- The abstract need to be improved
- Line 44; please replace seed plants by Spermaphytes
- Line 84 to 85; please rephrase the paragraph
- The main problem of this MSI is the relationship between metal tolerance and cell plasmolysis. I think the use of plasmolysis as a criterion of tolerance is not true. In fact, this phenomenon is a result of water leakage from the inside of the cell to the outside due to an external hypertonic solution. Please add more details regarding this topic
Comments on the Quality of English LanguageMinor editing of English language required
Reviewer 3 Report
Comments and Suggestions for Authors
Although the increasing pollution in the environment calls for the precise determination of metal accumulation in plants, I cannot recommend publishing this article because it is too descriptive and lacks mechanism analysis. Meanwhile, the experimental design is also somewhat unreasonable in this article. Especially since the author did not explain why they chose CuCl2, MnCl2, FeCl2, and Sb-acetate rather than other heavy metals. In addition, the combination of heavy metal concentrations also lacks scientific validity.